# Monitoring Redeployment-Associated Burnout in Healthcare Workers: A Real-Time Approach Using Ecological Momentary Assessment

**DOI:** 10.3390/healthcare13243217

**Published:** 2025-12-09

**Authors:** Abdulaziz Alkattan, Allison A. Norful, Cynthia X. Pan, Phyllis August, Robert S. Crupi, Joseph E. Schwartz, Andrew Miele, Elizabeth Brondolo

**Affiliations:** 1Department of Biomedical Informatics, Columbia University, New York, NY 10032, USA; azizalkattan@gmail.com; 2School of Nursing, Columbia University, New York, NY 10032, USA; aan2139@cumc.columbia.edu; 3Department of Medicine, NewYork-Presbyterian Queens, Flushing, NY 11355, USA; cxp9001@nyp.org; 4Weill Cornell Medicine, New York, NY 10021, USA; 5Lang Center for Research and Education, NewYork-Presbyterian Queens, Flushing, NY 11355, USA; paugust@med.cornell.edu; 6Department of Medicine, Weill Cornell Medicine, New York, NY 10021, USA; 7Department of Medicine, Division of Geriatric Medicine and Palliative Care, NewYork-Presbyterian Queens, Flushing, NY 11355, USA; roc9149@nyp.org; 8Department of Psychiatry and Behavioral Health, Renaissance School of Medicine, Stony Brook University, Stony Brook, NY 11794, USA; joseph.schwartz@stonybrookmedicine.edu; 9Research Education & Innovation, Medisys Health Network, New York, NY 11418, USA; andrew.miele18@my.stjohns.edu; 10Department of Psychology, St. John’s University, Queens, NY 11439, USA; 11Department of Family Medicine, Jamaica Hospital Medical Center, Jamaica, NY 11418, USA

**Keywords:** burnout, professional, resilience, psychological, ecological momentary assessment, COVID-19

## Abstract

**Highlights:**

**Abstract:**

**Background/Objectives**: Ecological momentary assessment (EMA) is a methodology that offers a real-time approach to monitoring clinician well-being, but its utility during high-intensity operational periods remains underexplored. This study examines the feasibility and performance of an EMA-based system for tracking clinical responsibilities and burnout among healthcare workers during the first year of the COVID-19 pandemic. **Methods**: Utilizing an intensive longitudinal design, 398 healthcare workers, including physicians, physician assistants, nurses, and trainees, completed brief EMA surveys every five days from April 2020 to March 2021. Burnout was assessed with a validated single-item measure and analyzed in relation to redeployment status and hospital caseloads. **Results**: The EMA approach successfully captured meaningful temporal fluctuations in burnout. Redeployment was associated with higher burnout levels (b = 0.125; *p* = 0.01), and rising caseloads amplified this effect (interaction b = 0.169; *p* = 0.001). Nurses showed the strongest caseload-related increases in burnout (b = 0.359; *p* < 0.001). These patterns persisted even after individuals returned to their usual roles. **Conclusions**: This study demonstrates that EMA is a scalable and sensitive approach for continuous burnout surveillance, capable of detecting role-specific and context-dependent stress responses in real time. EMA-based monitoring can support early identification of at-risk groups, guide staffing and redeployment decisions, and inform timely organizational interventions during crises and other periods of operational strain.

## 1. Introduction

Burnout, a syndrome characterized by emotional exhaustion, depersonalization, and reduced personal accomplishment, is a well-documented issue among healthcare workers [1,2]. Even before the COVID-19 pandemic, rates of burnout were rising, with substantial data indicating the COVID-19 pandemic accelerated burnout [3,4,5,6]. Burnout has been linked to decreased quality of care, increased medical errors, and organizational inefficiencies [2,7,8,9].

Occupational health frameworks, such as the Job Demands–Resources model or the Effort–Reward Imbalance model, demonstrate how factors including workload intensity, role clarity, and organizational support can shape burnout risk in healthcare environments [10,11,12]. Such frameworks underscore the influence of structural labor conditions on clinician well-being, but these factors can shift rapidly in the case of crises, making them difficult to track through traditional surveys. This highlights the need for approaches capable of capturing the effects of such changes in real time.

In response to surging patient volumes during the pandemic, hospitals implemented strategies such as redeployment of staff and increased nursing-to-patient ratios to address workforce shortages [13,14]. Redeployment is defined as shifting staff to unfamiliar roles, often with minimal notice and limited training [15,16,17,18]. Research has shown that redeployment during the COVID-19 pandemic was associated with increased burnout among healthcare workers, as well as heightened emotional exhaustion, anxiety, and depressive symptoms [3,4,5,6].

Monitoring the true prevalence and trajectory of burnout is difficult, particularly when working conditions shift rapidly. Ecological momentary assessment (EMA) is a methodological approach in which brief surveys or assessments are delivered repeatedly over time, often multiple times per day or week, to capture individuals’ thoughts, emotions, and behaviors as they occur in real-world settings. Originally developed for studies in ecological and behavioral psychology, EMA minimizes recall bias, increases ecological validity, and provides granular temporal data that traditional cross-sectional surveys cannot capture [19,20]. These features make EMA especially valuable in dynamic, high-pressure environments such as hospitals during public health emergencies, where emotional states and workload strain can change quickly [21,22,23,24].

Despite its strengths, EMA remains underutilized in studies of the healthcare workforce. Most existing research on burnout has relied on cross-sectional or short-term longitudinal designs, which fail to capture the nuanced and evolving interplay of factors like redeployment, workload, and the availability of support over extended periods [8,21,25]. Additionally, many studies focus on the immediate impacts of crises, neglecting the sustained effects of stress and burnout. This creates critical gaps in understanding burnout trajectories both during and after public health emergencies, knowledge essential for designing targeted interventions for high-risk groups. In contrast, EMA methods provide frequent, brief assessments to evaluate workforce responses to evolving workplace stressors.

In this analysis, we build on our original intensive longitudinal study using EMA methodology, which examined burnout trajectories among clinicians during the first year of the COVID-19 pandemic [21,26]. We focus on examining real-time fluctuations in burnout associated with changes in redeployment and caseload during and after the acute crisis periods. This permits evaluation of the practical value of EMA as a scalable tool for early risk identification, operational decision-making, and strengthening workforce resilience.

## 2. Materials and Methods

### 2.1. Study Design

This observational study employed ecological momentary assessment methodology in an intensive longitudinal survey design to examine the utility of real-time burnout monitoring among healthcare workers during the first year of the COVID-19 pandemic at a single healthcare center. This analysis represents a planned follow-up to our original study, which used the same EMA protocol to examine overall burnout trajectories among clinicians [21,26]. In the present secondary analysis, we focus specifically on how EMA captures short- and long-term fluctuations in burnout in relation to redeployment status and caseload variation. Redeployment was defined as assignment to a role or location outside the participant’s typical scope of work [3,4].

#### 2.1.1. Data Collection and Survey

Eligible participants included all frontline healthcare providers including physicians, physician assistants, nurses, and trainees. There were no specific exclusion criteria. Participants were recruited via hospital-wide email invitations sent to all clinical staff.

Surveys were administered every five days from 14 April 2020–31 March 2021, resulting in 70 assessments of redeployment status and burnout. The five-day interval was selected to balance temporal resolution with participant burden. Seven questions assessing demographic and professional information were administered in the initial survey. Subsequently, a 10-item EMA survey instrument, designed to be brief and simple, measured key dimensions of burnout, including emotional exhaustion, dimensions of self-efficacy, and workday responsibilities.

The survey took approximately one minute to complete. The Qualtrics platform was used to email surveys to clinical staff hospital-wide. Participants received daily reminders to complete the initial survey and two reminders for each additional survey. Full details of the survey instrument and the data collection procedures, including question structure, delivery schedule, and operational workflows are published in Pan et al. [21].

#### 2.1.2. Redeployment Status

To assess redeployment status, a single item asked if participants had worked in a clinical capacity within the past 24 h and was labeled “Redeployment Status”. We created two additional variables. On an observation level, we created a new variable labeled “Pre vs. Post Redeployment”, which permitted us to identify before and after redeployment periods. At a person level, a variable “Redeployment Group” was coded if participants were ever redeployed over the course of the study period.

#### 2.1.3. Burnout

Burnout was assessed with the Dolan et al. (2015) single-item-validated measure of burnout in healthcare workers [27]. Participants chose from 5 answers with responses ranging from no burnout to potentially debilitating burnout [28].

#### 2.1.4. Caseload and Case Severity

Hospital administration provided information about the daily Department of Medicine census, COVID-19-related caseload, and the number of COVID-19 patients receiving critical care. Caseloads were calculated using the average number of COVID-19 cases over a rolling period of five days (Caseloads were calculated using PROC EXPAND in SAS software, version 9.4 (SAS Institute Inc., Cary, NC, USA).

#### 2.1.5. Participant Comments

In addition to the quantitative survey, the participants were asked if they had any additional comments and given the option to provide a written text response. These comments were subject to content analysis to identify common themes. The aim was to help provide a richer context to interpret findings about relations among COVID-19 pandemic workplace demands, redeployment, and resultant burnout among individuals with different professional roles.

### 2.2. Data Analysis-Quantitative Survey Data

We conducted a multilevel mixed semi-linear regression analyses (MMLM) using PROC GLIMMIX in SAS software, version 9.4 (SAS Institute Inc., Cary, NC, USA) to examine how professional role, caseload, and redeployment were associated with burnout. Covariates included age, gender, professional role, and caseload based on established associations in prior burnout literature [18]. The current analysis included these variables as initial covariates. Additional details about the analytic approach, including model specifications and transformations, are available in Pan et al. [21]. Predictors in the model included redeployment status, caseload, and professional role, along with their interactions. The final model included significant effects only and can be found in Appendix A.

### 2.3. Data Analysis-Comments

A total of 479 comments from 143 participants were collected over the course of the study. Although these were optional, unstructured comments, our team decided to use the data to look for prevailing themes and trends, specifically around redeployment and burnout. We did not obtain permission to publish the comments verbatim, so references to comments in this paper do not include direct quotations of full responses. Using content analytic approaches that we have used previously, each comment was categorized by its main topics [29]. An initial codebook was developed based on recurring concepts which were identified in a subset of comments. Coding continued until thematic saturation was reached, indicated by no new themes emerging. To ensure the accuracy of our analysis, two team members separately reviewed all comments. Discrepancies were resolved through discussion until consensus was reached. This content-analytic approach follows methodologies used in previously published research [30].

## 3. Results

### 3.1. Characteristics of the Sample

This study utilizes the same study cohort as our previous publication [21,26]. A total of 398 participants completed the initial survey. Almost 81% (*n* = 322) completed the survey multiple times. On average, participants completed 12 surveys (median = 5), yielding a total of 5070 surveys over one year. Of the 398 participants, 94% (*n* = 372) provided data on burnout.

Redeployment status and burnout were assessed at each EMA survey. There were no significant associations between the number of surveys a participant completed and their burnout scores. There was no significant difference between participants completing the survey more than once and those who completed the survey only once on any demographic or professional role variables [21].

Nursing and nurse practitioners comprised the largest group in the sample (55.6%, *n* = 221), followed by attending physicians (22%, *n* = 87), trainees (residents and fellows) (15%, *n* = 60), and physician assistants (11%, *n* = 44). The study population was primarily self-identified women (70%, *n* = 277), with 79% (*n* = 306) under the age of fifty. Demographically, the study population mirrored the institute’s workforce diversity. The majority identified as Asian (44.3%, *n* = 171), followed by White (37.5%, *n* = 145), Black (10.4%, *n* = 40), and Latino (7.8%, *n* = 30). Most participants had less than five years of experience (41.2%, *n* = 164).

### 3.2. Redeployment

Of the 398 participants, 58.52% were redeployed during at least one of their survey responses (*n* = 230) and 41.48% were never redeployed (*n* = 163) and 5 were missing redeployment data. About 10% of participants were redeployed during all survey responses (*n* = 37). There were no statistically significant differences in age (*p* = 0.86), gender (*p*= 0.59), professional role (*p* = 0.17) or hospital department (*p* = 0.62) between those who were ever redeployed and those who were never redeployed.

### 3.3. Inferential Analyses of Workplace Predictors of Burnout

The final model obtained from multilevel mixed semi-linear regression analyses estimating the relationship between redeployment and burnout is shown in Appendix A.

#### 3.3.1. Role-Specific Burnout Trends

As shown in Figure 1, the relationship between burnout and caseload differed significantly by both redeployment status and professional role. Nurses experienced a significant increase in burnout as caseloads rose (b = 0.3591; SE = 0.0746; *p* < 0.0001), an effect not observed in other professional roles. The estimates and standard errors used to generate this figure are presented in Appendix A
Table A1.

#### 3.3.2. Interactions of Caseload and Redeployment

Caseload moderated the effect of redeployment on burnout. As caseloads increased, the positive relation between redeployment and burnout increased (b = 0.1691; SE = 0.0491; *p* = 0.001). The combination of high caseloads and redeployment was associated with the most pronounced burnout levels across all participants. This effect did not vary by professional role.

#### 3.3.3. Long-Term Associations of Redeployment to Burnout

Burnout levels were significantly higher after participants’ initial redeployment compared to the period before redeployment (b = 0.1251; SE = 0.0488; *p* = 0.010). This effect persisted regardless of whether participants returned to their usual roles and was consistent across all professional roles.

### 3.4. Analysis of Comments

During the data collection period, approximately one-third of participants (35.9%, *n* = 143) provided at least one comment, resulting in a total of 479 comments. These comments provide qualitative evidence of the stressors captured by the quantitative components of the EMA tool, offering deeper insights into the lived experiences driving burnout. For example, participants described feeling “uncomfortable” and “horrible” when thrust into unfamiliar roles without notice. Regarding the lack of training and guidance, participants expressed feeling “outside our comfort zone” and called for in-service and/or training to familiarize them with equipment, policies, and protocols.

The most prevalent themes are outlined in Table 1, with 17.75% (*n* = 85) of all comments directly addressing redeployment-related concerns. Analysis of redeployment-related comments highlighted several critical challenges that clinicians experienced during redeployment. These included:Being reassigned to unfamiliar roles with minimal notice.Insufficient training for redeployed responsibilities.Lack of guidance regarding policies and procedures on new units.Concerns about personal safety in less familiar environments.
healthcare-13-03217-t001_Table 1Table 1Percentage of select themes found in responses to the additional comments question.ThemePercent of Comments (%)Increased stress27.35%Lack of support24.63%Poor staffing20.87%Discontent with redeployment17.75%Feelings of burnout13.57%


## 4. Discussion

The analyses from this study demonstrate the utility of ecological momentary assessment (EMA) as an innovative and scalable tool for real-time monitoring of burnout among healthcare workers during crises. These brief surveys offer a practical and user-friendly solution for real-time monitoring, requiring minimal time investment from participants while delivering more nuanced and actionable insights compared to traditional retrospective surveys. Our use of EMA enabled us to capture dynamic, temporal variations in burnout associated with changing workplace conditions in frontline healthcare workers during the first year of the COVID-19 pandemic. The longitudinal design allowed for the identification of critical patterns, such as the persistent effects of redeployment and the amplifying role of increased caseloads on burnout levels. These findings highlight the power of EMA to deliver actionable data that can guide organizational decision-making.

Participants who experienced redeployment reported higher burnout levels than those who remained in their usual roles, with these effects persisting over time and intensifying when caseloads increased. Redeployment-related burnout often did not resolve even after individuals returned to their original roles, suggesting that redeployment represents a significant stressor with potential long-term impacts [31,32]. Notably, nurses demonstrated the strongest burnout response to surging caseloads, which may reflect their continuous bedside responsibilities, disproportionate emotional labor, and central role in implementing rapidly changing clinical protocols during the pandemic [32]. These overlapping pressures likely contributed to greater cumulative strain among nurses compared to other professional groups.

EMA’s ability to capture these patterns in near real-time offers organizations a critical advantage, enabling early identification of at-risk staff and the implementation of targeted interventions before burnout becomes entrenched, as supported by existing literature [33]. These data were distributed to all administrators and staff in newsletters distributed throughout the pandemic. EMA also allowed for the collection of qualitative feedback from participants, which provided deeper context for the observed quantitative findings. Comments highlighted recurring themes, such as the stress of transitioning to new roles with minimal preparation, the lack of formal guidance on new policies and procedures, and concerns for personal safety in unfamiliar environments. These lived experiences underscore the importance of integrating real-time qualitative insights into burnout monitoring efforts, as they can reveal actionable gaps in organizational processes and staff support systems.

## 5. Limitations

This study has several limitations. The initial response rate of approximately 20%, while consistent with similar studies involving voluntary clinician participation, was accompanied by participant attrition over the study period [7]. This declining response rate could affect the reliability of longitudinal findings. Despite its strengths, the use of EMA is not without challenges, including the potential for survey fatigue among participants due to frequent data collection. Further, the nature of EMA data creates large datasets that require advanced statistical methods and specialized expertise to analyze effectively. However, packages are available from SAS, such as PROC MIXED and PROC GLIMMIX in SAS version 9.4 (SAS Institute Inc., Cary, NC, USA), R (R Foundation for Statistical Computing, Vienna, Austria), and other analytic software which facilitate these analyses. The reliance on brief, repeated assessments can limit the depth and richness of data compared with traditional qualitative approaches such as interviews or focus groups.

The single-site design and convenience sampling method may restrict the generalizability of the results of the survey to other healthcare systems or disaster scenarios. Additionally, the absence of baseline burnout data collected prior to the pandemic limits our ability to fully attribute observed burnout trends to the pandemic itself. Finally, while the inclusion of participant comments adds depth to the study, a more robust qualitative approach, such as structured interviews or focus groups, would provide additional context and enhanced understanding of the lived experiences of healthcare workers.

Future research should explore the application of EMA informed interventions across diverse healthcare settings and crisis scenarios, assessing its utility in contexts such as natural disasters or prolonged humanitarian emergencies. Additionally, studies should investigate the long-term effectiveness of EMA-informed interventions in reducing burnout and supporting workforce resilience over time. By leveraging the capabilities of EMA, healthcare systems can build more adaptive, responsive, and sustainable approaches to disaster preparedness and workforce management.

## 6. Conclusions

This study underscores the value of real-time monitoring tools such as ecological momentary assessment (EMA) for identifying and responding to burnout among healthcare workers during crisis conditions. The findings demonstrate that redeployment, while necessary in emergency scenarios, is associated with persistent increases in burnout, particularly when compounded by high caseloads. Importantly, EMA’s ability to capture nuanced, real-time data on burnout provided actionable insights to hospital administrators and managers into the temporal and contextual factors exacerbating stress among healthcare workers. These insights emphasize the need for proactive strategies to mitigate the immediate and long-term effects of redeployment and workload surges.

To enhance workforce resilience and ensure operational readiness during future crises, healthcare organizations should consider the following actionable recommendations:Adopt Real-Time Monitoring: Implement EMA or similar tools to continuously track staff well-being and burnout during emergencies. These tools enable the early identification of at-risk individuals and allow for dynamic, data-driven interventions tailored to evolving conditions.Develop Redeployment Protocols: Establish structured protocols that include preemptive training, clear role-specific guidance, and organized hand-off processes to reduce the stress of sudden redeployment, allowing for dynamic caseload adjustments during surges.Integrate Mental Health and Peer Support Programs: Provide accessible mental health resources and establish peer support networks to specifically address the emotional challenges associated with redeployment and high workloads. These programs should be sustained beyond crises to support long-term workforce well-being.Enhance Organizational Communication: Maintain consistent and transparent communication channels to disseminate updates on policies, expectations, and available resources. Clear communication fosters trust and reduces uncertainty during crises.Expand Professional Development Opportunities: Offer cross-training and professional development programs to prepare staff for redeployment scenarios. Building skills across roles can ease transitions during emergencies and reduce stress by fostering confidence in diverse responsibilities.

## Figures and Tables

**Figure 1 healthcare-13-03217-f001:**
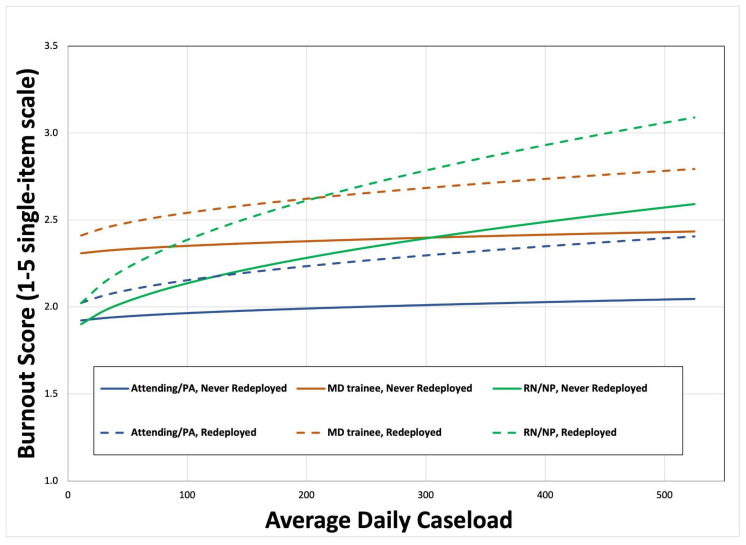
Relationship of burnout to caseload, with variation by redeployment status and professional role.

## Data Availability

Data are available on reasonable request from the corresponding author (E.B.) due to privacy and ethical restrictions. De-identified analytic code and variable definitions will be shared upon request. E.B. has full access to all data and takes responsibility for the integrity and accuracy of the analysis.

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
