# Peer review of "Monitoring Redeployment-Associated Burnout in Healthcare Workers: A Real-Time Approach Using Ecological Momentary Assessment"

_healthcare, 2025, doi:10.3390/healthcare13243217_

Round 1
Reviewer 1 Report
Comments and Suggestions for Authors
The abstract is structured; instead of "significant associations," consider adding a key statistic check the keywords in accordance with MeSH.
In the introduction, please include more examples of specific frameworks that assess labor practices in healthcare, in relation to scientific literature (for e.g. doi: 10.3390/medicina59040796 ).
The methods are well structured in subsections. The manuscript notes a declining response rate; a formal analysis comparing early dropouts with persistent participants on key demographics and baseline burnout scores would strengthen the case for the validity of the longitudinal findings against attrition bias. Please describe the coding process (e.g., inductive vs. deductive), how inter-rater reliability was ensured between the two coders, and how thematic saturation was assessed would bolster the rigor of the qualitative component.
The results are logically presented, and the key findings are impactful. The interaction plots (Figure 1) effectively visualize complex relationships. But please add in Figure 1 a clearer title and axis labels directly to the figure image.
In the discussions the result that nurses were most affected is highlighted but not sufficiently theorized; discuss potential reasons, such as nurses' continuous, intimate patient contact, their central role in operationalizing new protocols, or pre-existing systemic pressures that were amplified during the pandemic.
The conclusions are concise.
The references are adequate but should be extended as suggested above to reach a minimum of 30 given the type of paper.
Author Response
Reviewer 1
Comment 1: The abstract is structured; instead of “significant associations,” consider adding a key statistic. Check keywords in accordance with MeSH.
Response 1: Thank you for this suggestion. We revised the abstract to include specific statistical estimates reflecting the main associations, and refined the keywords to align with MeSH terminology.
Revisions made on p.2, Abstract, lines 48–50: Redeployment was associated with higher burnout levels (b = 0.125; p = .01), and rising caseloads amplified this effect (interaction b = 0.169; p = .001). Nurses showed the strongest caseload-related increases in burnout (b = 0.359; p < .001).”
Keywords revised: Burnout, Professional; Resilience, Psychological; Surveys and Questionnaires; COVID-19.
Comment 2: In the introduction, please include more examples of specific frameworks that assess labor practices in healthcare, in relation to scientific literature (e.g. doi:10.3390/medicina59040796).
Response 2: We agree. We have added explicit reference to occupational health frameworks (e.g., Job Demands–Resources; Effort–Reward Imbalance), and cited seminal papers as well as the Medicina article suggested.
Revisions on p.2, Introduction, paragraph 1, lines 65–67.
“Occupational health frameworks, such as the Job Demands–Resources model or the Effort–Reward Imbalance model, demonstrate how factors including workload intensity, role clarity, and organizational support can shape burnout risk in healthcare environments”
Comment 3: The manuscript notes a declining response rate; a formal analysis comparing early dropouts with persistent participants … would strengthen the case against attrition bias.
Response 3: We have clarified in the Results section that participants who completed only one survey did not differ demographically or professionally from repeat completers.
Language added on p.4, Sec 3.1, lines 198–199. “There were no significant associations between the number of surveys a participant completed and their burnout scores.” This is addressed in more detail in the initial publication on the study:
Pan CX, Crupi R, August P, et al. An intensive longitudinal assessment approach to surveilling trajectories of burnout over the first year of the COVID pandemic. Int J Environ Res Public Health. 2023;20(4):2930. doi:10.3390/ijerph20042930
Comment 4: Please describe the coding process, how inter-rater reliability was ensured, and how thematic saturation was assessed.
Response 4: We have expanded Section 2.4 to describe:
– development of initial codebook
– independent coding by two reviewers
– consensus resolution approach
– saturation criteria
Revisions located p.4, Sec 2.4, lines 178–190. We have also included a reference to an article where we have used similar methods
Comment 5: Add a clearer title and axis labels directly to the figure image.
Response 5: Completed. We revised Figure 1 to include clearer axes, explicit units, and a revised descriptive title.
Updated Figure on p.6.
Comment 6: In Discussion, the result that nurses were most affected is highlighted but not sufficiently theorized; discuss potential reasons.
Response 6: Expanded Discussion includes drivers such as proximity to patient care, emotional load, unit-level operational strain, and implementation burden.
“Notably, nurses demonstrated the strongest burnout response to surging caseloads, which may reflect their continuous bedside responsibilities, disproportionate emotional labor, and central role in implementing rapidly changing clinical protocols during the pandemic. These overlapping pressures likely contributed to greater cumulative strain among nurses compared to other professional groups.”
Added on p.7, Discussion, paragraph 2.
Comment 7: References should be extended to minimum 30.
Response 7: Completed. The revised manuscript now contains 31 references.
Updated in References section, p.10–11.
Reviewer 2 Report
Comments and Suggestions for Authors
The article fits within the thematic scope of the journal. Nevertheless, it requires some additional comments, clarifications, or corrections:
Lines 70–79: In my opinion, it is not clear what EMA is. Although the authors mention several functions of this tool, they do not explain the nature of EMA. I believe that EMA should be described in more detail: what the origins of this tool are (briefly), what its main features are—particularly those relevant to the conducted research—and what its essential functions are.
Lines 109–116: Were the studies conducted in multiple hospitals or only in one? I believe this should be clearly stated in the article. This information is important because, at the current stage, it is unclear whether the presented research can be considered representative.
Line 133: The text reads “23” but it should be “[23]”.
Line 196: The text reads “18” but it should be “[18]”.
Line 272: For the sake of clarity, I believe it would be better to create a separate section and call it “Limitations”.
Author Response
Comment 1: Lines 70–79: EMA is not clearly explained.
Response 1: Agree. We appreciated the reviewers’ communication about the need for greater explication. We expanded EMA definition, described historical origins, core features, and relevance to crisis contexts.
Updated p.2–3, Introduction paragraph 3. Lines 81-87
“Ecological momentary assessment (EMA) is a methodological approach in which brief surveys or assessments are delivered repeatedly over time, often multiple times per day or week, to capture individuals’ thoughts, emotions, and behaviors as they occur in real-world settings. Originally developed in ecological and behavioral psychology, EMA minimizes recall bias, increases ecological validity, and provides granular temporal data that traditional cross-sectional surveys cannot capture”
Comment 2: Clarify whether study conducted in one or multiple hospitals.
Response 2: We clarified that this study was conducted at a single academic medical center.
Location: p.3, Sec 2.1, lines 115.
Comment 3: Line 133 “23” should be “[23]”.
Response 3: Corrected.
Comment 4: Line 196 “18” should be “[18]”.
Response 4: Corrected.
Comment 5: Create a separate limitations section titled “Limitations.”
Response 5: Completed.
A standalone Section 5 “Limitations” has been added.
Reviewer 3 Report
Comments and Suggestions for Authors
The article “Monitoring Redeployment-Associated Burnout in Health Care Workers: A Real-Time Approach Using Ecological Momentary Assessment” addresses an important and timely topic; however, it presents several significant limitations in both structure and analytical depth. Although the study explores burnout across various professional roles within the hospital setting, it falls short by not comparing these groups or examining how they function in different clinical areas.
In addition, the methodology does not clearly describe the participant selection process, making it difficult to assess the validity and representativeness of the sample. The authors also fail to specify the statistical analyses performed, the tests selected, or the rationale behind their use, which limits the transparency and reproducibility of the study.
The tables, figures, and graphs presented do not meet the basic criteria required for manuscript publication and acceptance in this way. In addition, the authors do not explain how open-ended questions were normalized or how consensus was reached in the qualitative responses. Furthermore, the results do not include an adequate discussion of their applicability, reducing the interpretative value of the findings.
Although the topic is innovative, the presentation of results lacks refinement, and the analysis is largely limited to demographic data, without deeper insights that could enhance the scientific contribution of the work. Therefore, it is recommended that the authors restructure the manuscript to better highlight the importance of their findings within such a highly stressful environment as a hospital.
It would also be beneficial to compare the results with previous studies conducted in Latin America, Europe, and the United States, where burnout has been explored in hospital, academic, and educational contexts, thereby situating the study within a stronger international framework.
Finally, the references should be updated and formatted according to the journal’s guidelines.
Author Response
Comment 1: Study explores burnout across roles but does not compare groups.
Response 1: We appreciate the need for greater clarity. We have improved the legibility of Figure one in which we present differences among professional role groups in burnout.
Comment 2: Participant selection process unclear.
Comment 3: Statistical analyses, tests, and rationale insufficiently described.
Response 2,3: Thank you for this comment. We realize that we were not explicit enough in saying this was a follow-up analysis. As this is a secondary analysis of this data set focusing on the utility of the tool, information on the group comparisons, participation selection, and statistical analysis is presented in the initial publications, including
Pan CX, Crupi R, August P, et al. An intensive longitudinal assessment approach to surveilling trajectories of burnout over the first year of the COVID pandemic. Int J Environ Res Public Health. 2023;20(4):2930. doi:10.3390/ijerph20042930
Ju TR, Mikrut EE, Spinelli A, et al. Factors associated with burnout among resident physicians responding to the COVID-19 pandemic: a 2-month longitudinal observation study. Int J Environ Res Public Health. 2022;19(15):9714. doi:10.3390/ijerph19159714
Comment 4: Tables, figures, graphs do not meet criteria.
Response 4: Figure 1 has been redesigned with clearer labeling, legend, and axis scaling.
Updated p.6.
Comment 5: Open-ended comments unstructured; need normalization & consensus.
Response 5: We added a detailed coding protocol incl.:
– codebook development
– double coding
– saturation definition
– consensus process
p.4, Sec 2.4, lines 5–21.
Comment 6: Discussion did not adequately address applicability.
Response 6: We added text linking EMA utility to operational surveillance, incident command utility, and targeted workforce protection strategies. We also discussed the impact and rationale specifically for nursing.
“Notably, nurses demonstrated the strongest burnout response to surging caseloads, which may reflect their continuous bedside responsibilities, disproportionate emotional labor, and central role in implementing rapidly changing clinical protocols during the pandemic. These overlapping pressures likely contributed to greater cumulative strain among nurses compared to other professional groups.” p.8, Discussion, paragraph 2, lines 321-328.
We also have provided explicit recommendations based on the data on P.8, conclusion, lines 350 to 370.
Comment 7: Compare with prior studies internationally.
Response 7: in the introduction, comparative papers looking at similar outcomes but not using EMA are mentioned, including:
- Dionisi T, Sestito L, Tarli C, et al. Risk of burnout and stress in physicians working in a COVID team: a longitudinal survey. Int J Clin Pract. 2021;75(11):e14755. doi:10.1111/ijcp.14755 (Italy)
- Satomi E, Souza PMR, Thomé BDC, et al. Fair allocation of scarce medical resources during COVID-19 pandemic: ethical considerations. Einstein (Sao Paulo). 2020;18:eAE5775. doi:10.31744/einstein_journal/2020AE5775 (Brazil)
Comment 8: References should be updated & reformatted.
Response 8: Completed. Reference list expanded and reformatted to MDPI reference style.
Round 2
Reviewer 2 Report
Comments and Suggestions for Authors
I have no more comments.
Author Response
Thank you
Reviewer 3 Report
Comments and Suggestions for Authors
Although the authors have provided a synthesis of the previously raised comments, several important issues still require attention.
First, the Highlights section needs to be reformulated. In its current form, it reads almost as a direct copy of the Abstract, which should be avoided. The Highlights should briefly emphasize the novelty and key contributions of the study, not restate the abstract.
Second, Figure 1 appears very opaque or low-quality, making it difficult to distinguish its elements. The figure should be improved to ensure clarity and readability.
A particularly important issue concerns the table presented in the Appendix titled “Mixed Model Examining Effects of Professional, Redeployment Status at Time of Survey, Redeployment Status Overall on Burnout.” This table is never mentioned in the main document, nor is it discussed as it should be. Moreover, several values in the table contain asterisks, yet no explanation or legend is provided to clarify their meaning. The authors should carefully evaluate whether this table should remain; if so, it must be properly integrated, referenced, explained, and discussed. Otherwise, it should be removed.
Additionally, in the Results section the authors use expressions such as “As we have reported in the initial paper,” yet no other paper is cited. This phrasing generates confusion and must be corrected. I strongly recommend having this section reviewed by an expert in English grammar or restructuring it to ensure proper syntax and clarity.
Similarly, in the Discussion section expressions like “Our prior research and these analyses demonstrate” continue to appear, but more precise and appropriate phrasing should be considered to strengthen the narrative.
Overall, the manuscript still requires further polishing to improve clarity, structure, and coherence.
Author Response
Comment 1: First, the Highlights section needs to be reformulated. In its current form, it reads almost as a direct copy of the Abstract, which should be avoided. The Highlights should briefly emphasize the novelty and key contributions of the study, not restate the abstract.
Response1: Thank you for the comment. We have re-written the highlights to emphasize the novelty and key contributions and have differentiated that from the abstract.
"Highlights
What are the main findings?
- Ecological momentary assessment (EMA) methods can capture dynamic trends in clinician burnout during a prolonged hospital crisis.
- Redeployment and higher caseloads increased burnout, with effects persisting after return to usual roles; nurses were most affected.
What is the implication of the main finding?
- Redeployment is a commonly used strategy to handle increased needs for personnel during prolonged hospital crises. However, the data suggest that redeployment presents risks for persistent burnout under some conditions and for some personnel. Preventative efforts to reduce these risks are needed.
- This novel application of EMA, using a low-intensity continuous surveillance method, can provide actionable information in real time to guide the deployment of burnout-mitigation efforts and to permit better targeting of these efforts to specific time periods and high-risk groups. "
Comment 2: Second, Figure 1 appears very opaque or low-quality, making it difficult to distinguish its elements. The figure should be improved to ensure clarity and readability.
Response 2: The colors and contrast of the figure have been improved, and a PDF version has also been submitted
Comment 3: A particularly important issue concerns the table presented in the Appendix titled “Mixed Model Examining Effects of Professional, Redeployment Status at Time of Survey, Redeployment Status Overall on Burnout.” This table is never mentioned in the main document, nor is it discussed as it should be. Moreover, several values in the table contain asterisks, yet no explanation or legend is provided to clarify their meaning. The authors should carefully evaluate whether this table should remain; if so, it must be properly integrated, referenced, explained, and discussed. Otherwise, it should be removed.
Response 3: Thank you for identifying this issue. We agree that the appendix table required clearer integration into the manuscript. Because the table provides transparency regarding the model estimates and supports the inferential results presented in the text, we decided to retain it and have revised the manuscript accordingly.
- A label was added: "Note: * p < 0.05. Estimates marked with an asterisk indicate statistically significant effects."
-
In Section 2.2, we now note that the final model output is shown in Appendix A and underlies all inferential findings.
-
In Section 3.4 and 3.4.1, we added direct references to Appendix A to indicate where the full model estimates and corresponding effect sizes can be found.
- "As shown in Figure 1, the relationship between burnout and caseload differed significantly by both redeployment status and professional role. Nurses experienced a significant increase in burnout as caseloads rose (b = 0.3591; SE = 0.0746; P < .0001), an effect not observed in other professional roles. The estimates and standard errors used to generate this figure are presented in Table 1A, Appendix A. "
Comment 4: Additionally, in the Results section the authors use expressions such as “As we have reported in the initial paper,” yet no other paper is cited. This phrasing generates confusion and must be corrected. I strongly recommend having this section reviewed by an expert in English grammar or restructuring it to ensure proper syntax and clarity.
Similarly, in the Discussion section expressions like “Our prior research and these analyses demonstrate” continue to appear, but more precise and appropriate phrasing should be considered to strengthen the narrative.
Response 4: Thank you for noting this source of confusion. We have revised the wording in our results and discussion, particularly when referring to our previous work. We maintain the reference in section 3.1 to indicate that the same cohort was used for these analyses.
Results: "This study utilizes the same study cohort as our previous publication [18,30]."
In the discussion, we removed references to our prior work as it does not pertain to the new findings.
Discussion: "The analyses from this study demonstrate the utility of ecological momentary assessment (EMA) as an innovative and scalable tool for real-time monitoring of burnout among health care workers during crises."